# Replication Study: Inhibition of BET recruitment to chromatin as an effective treatment for MLL-fusion leukaemia

Xiaochuan Shan[1], Juan Jose Fung[2], Alan Kosaka[2], Gwenn Danet-Desnoyers[1], Reproducibility Project: Cancer Biology*

[1]University of Pennsylvania, Perelman School of Medicine, Stem Cell and Xenograft Core, Philadelphia, United States; [2]ProNovus Bioscience, LLC, Mountain View, United States

*For correspondence: tim@cos.io; nicole@scienceexchange.com

Group author details:
Reproducibility Project: Cancer Biology See page 14

**Abstract** In 2015, as part of the Reproducibility Project: Cancer Biology, we published a Registered Report (Fung et al., 2015), that described how we intended to replicate selected experiments from the paper "Inhibition of BET recruitment to chromatin as an effective treatment for MLL-fusion leukaemia" (Dawson et al., 2011). Here, we report the results of those experiments. We found treatment of MLL-fusion leukaemia cells (MV4;11 cell line) with the BET bromodomain inhibitor I-BET151 resulted in selective growth inhibition, whereas treatment of leukaemia cells harboring a different oncogenic driver (K-562 cell line) did not result in selective growth inhibition; this is similar to the findings reported in the original study (Figure 2A and Supplementary Figure 11A,B; Dawson et al., 2011). Further, I-BET151 resulted in a statistically significant decrease in *BCL2* expression in MV4;11 cells, but not in K-562 cells; again this is similar to the findings reported in the original study (Figure 3D; Dawson et al., 2011). We did not find a statistically significant difference in survival when testing I-BET151 efficacy in a disseminated xenograft MLL mouse model, whereas the original study reported increased survival in I-BET151 treated mice compared to vehicle control (Figure 4B,D; Dawson et al., 2011). Differences between the original study and this replication attempt, such as different conditioning regimens and I-BET151 doses, are factors that might have influenced the outcome. We also found I-BET151 treatment resulted in a lower median disease burden compared to vehicle control in all tissues analyzed, similar to the example reported in the original study (Supplementary Figure 16A; Dawson et al., 2011). Finally, we report meta-analyses for each result.

## Introduction

The Reproducibility Project: Cancer Biology (RP:CB) is a collaboration between the Center for Open Science and Science Exchange that seeks to address concerns about reproducibility in scientific research by conducting replications of selected experiments from a number of high-profile papers in the field of cancer biology (*Errington et al., 2014*). For each of these papers a Registered Report detailing the proposed experimental designs and protocols for the replications was peer reviewed and published prior to data collection. The present paper is a Replication Study that reports the results of the replication experiments detailed in the Registered Report (*Fung et al., 2015*), for a paper by Dawson et al., and uses a number of approaches to compare the outcomes of the original experiments and the replications.

In 2011, Dawson et al. demonstrated targeting BET proteins as an effective strategy for modulating aberrant gene expression associated with mixed-lineage leukaemia (MLL), a genetically distinct form of acute leukaemia. Selective inhibition of cell viability and *BCL2* expression in leukaemia cells

harboring MLL fusions was observed with the BET bromodomain inhibitor, I-BET151, in contrast to leukaemia cells with alternate oncogenic drivers (*Dawson et al., 2011*). Furthermore, efficacy of I-BET151 was tested in a xenograft model of MLL, which resulted in a statistically significant increase in survival compared to vehicle control treated animals.

The Registered Report for the paper by Dawson et al. described the experiments to be replicated (Figures 2A, 3D, 4B and D, and Supplementary Figures 11A-B and 16A), and summarized the current evidence for these findings (*Fung et al., 2015*). Recent studies have investigated the efficacy of targeting BET bromodomains in other cancer types. Studies using structurally distinct BET inhibitors, JQ1 and OTX015, have reported these inhibitors to be highly active in various cell lines, mouse models, and primary patient samples of MLL and other types of AML (*Chen et al., 2013*; *Coudé et al., 2015*; *Fiskus et al., 2014*; *Herrmann et al., 2013*; *Mertz et al., 2011*; *Zuber et al., 2011*). Furthermore, I-BET151 was reported to be active against AML with mutations involving the nucleophosmin (*NPM1*) gene (NPM1c AML) (*Dawson et al., 2014*). Additional studies examining the antitumor effects of BET bromodomain inhibition include other cancer types, such as gastric cancer (*Montenegro et al., 2016*), childhood sarcoma (*Bid et al., 2016*), triple negative breast cancer (*da Motta et al., 2017*; *Shu et al., 2016*), and ovarian cancer (*Zhang et al., 2016*). Additionally, evidence indicating potential acquired resistance to BET inhibitors has been reported (*Fong et al., 2015*; *Kumar et al., 2015*; *Rathert et al., 2015*), along with studies which have found that treatment with combinatorial drugs can mitigate acquired resistance (*Asangani et al., 2016*; *Kurimchak et al., 2016*; *Yao et al., 2015*). Furthermore, a recent study reported synergistic effects of combinatorial treatment with a disruptor of telomeric silencing 1-like (DOT1L) inhibitor and I-BET151 in MLL cells and mouse leukaemia models (*Gilan et al., 2016*). Studies have also addressed the nonclinical safety of BET inhibition and found that BET bromodomain inhibition can impact normal intestinal homeostasis and repair (*Nakagawa et al., 2016*). There are several variations of I-BET bromodomain inhibitors currently in clinical trials (*Chaidos et al., 2015*; *French, 2016*; *Wadhwa and Nicolaides, 2016*). A phase I trial to determine the recommended dose of BET inhibitor OTX015/MK-8628 reported that although the inhibitor was found to be tolerated, toxic effects such as thrombocytopenia were commonly observed (*Amorim et al., 2016*). An additional clinical study testing BET inhibition with OTX015/MK-8628 in four patients with advanced NUT midline carcinoma, with confirmed BRD4-NUT fusions, found early clinical benefits reported for two patients with one achieving post-treatment disease stabilization (*Stathis et al., 2016*).

The outcome measures reported in this Replication Study will be aggregated with those from the other Replication Studies to create a dataset that will be examined to provide evidence about reproducibility of cancer biology research, and to identify factors that influence reproducibility more generally.

## Results and discussion

### Evaluating the selective inhibition of a MLL-fusion leukaemic cell line with I-BET151

Using the same BET bromodomain inhibitor as Dawson et al., we aimed to independently replicate an experiment analyzing the ability of I-BET151 to selectively inhibit the growth of the MLL-fusion leukaemic cell line MV4;11 (MLL-AF4 fusion). Similar to the original study, the K-562 leukaemic cell line, which is oncogenically driven by tyrosine kinase activation, served as a negative control. This experiment is comparable to what was reported in Figure 2A and Supplementary Figure 11A-B of *Dawson et al. (2011)* and described in Protocols 1–2 in the Registered Report (*Fung et al., 2015*). While the original study included a variety of human leukaemia cell lines, this replication attempt was restricted to the MV4;11 and K-562 cell lines, which were utilized in further experiments. For each cell line, cellular $IC_{50}$ values were determined (*Figure 1*, *Figure 1—figure supplement 1*, *Figure 1—figure supplement 2*) with a mean absolute $IC_{50}$ value of 9.03 nM [n = 5, *SD* = 8.27 nM] for I-BET151 treated MV4;11 cells, while the K-562 $IC_{50}$ estimates were not able to be accurately determined following published guidelines (*Sebaugh, 2011*). This compares to the original study that reported an $IC_{50}$ value of 26 nM for MV4;11 cells, with the K-562 $IC_{50}$ value also unable to be accurately determined (*Dawson et al., 2011*).

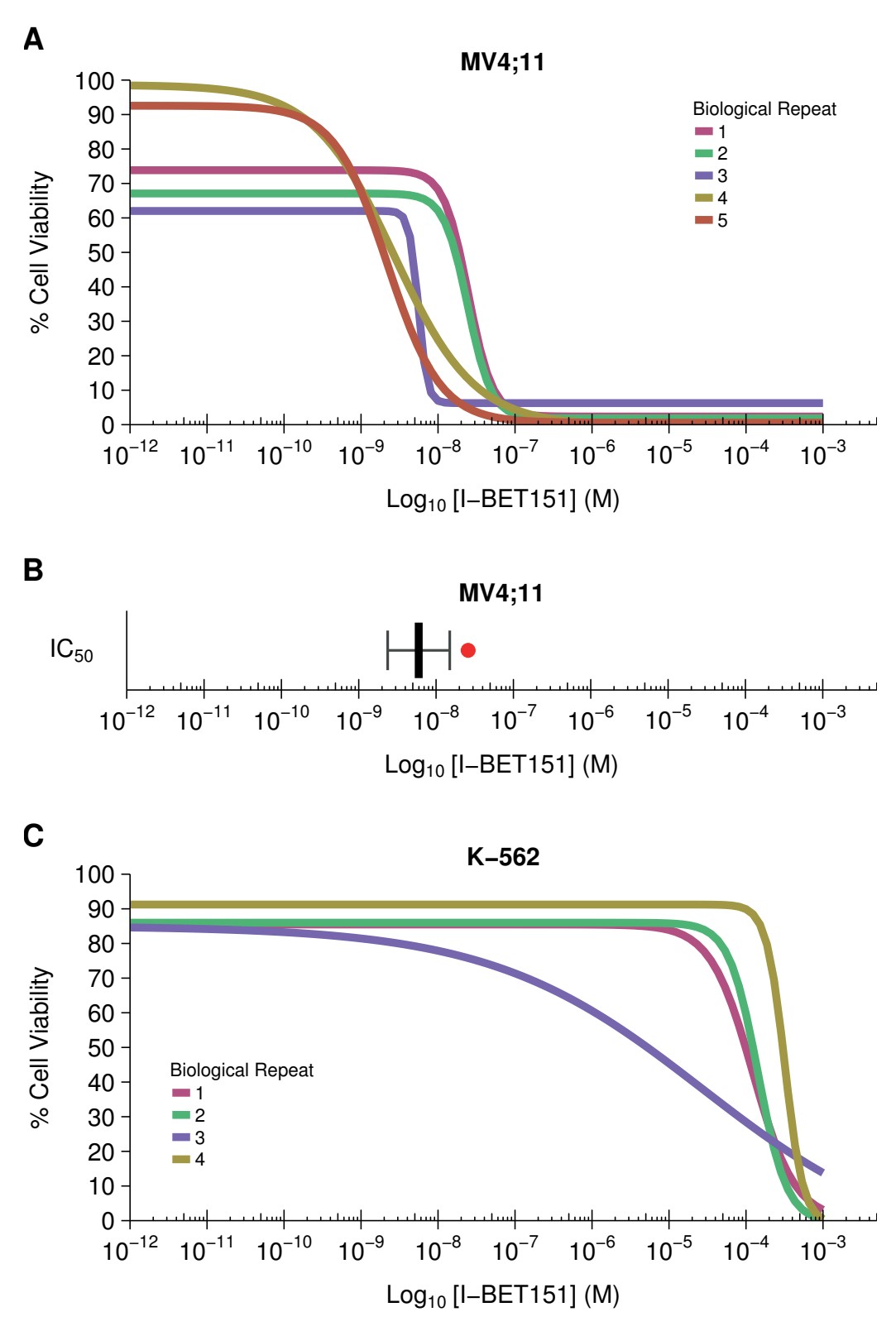

**Figure 1.** Cellular I-BET151 dose response curves in MLL-fusion leukaemia and non-MLL-fusion leukaemia cell lines. Cell viability assays were performed for a MLL-fusion leukaemia cell line (MV4;11) and a non-MLL-fusion leukaemia cell line (K-562) with the I-BET151 inhibitor. Curves and absolute $IC_{50}$ values were determined for each biological repeat. (**A**) Dose response curves for each biological repeat [n = 5] for the MV4;11 cell line. Percent cell viability is relative to DMSO treated cells. I-BET151 doses range from 1 pM to 1 mM. Exploratory one-sample *t*-test of $IC_{50}$ values compared to a

*Figure 1 continued on next page*

*Figure 1 continued*

constant of 100 µM; $t(4)$ = 20.63, $p$=3.26×$10^{-5}$; Cohen's $d$ = 9.23, 95% CI [3.13, 15.45]. (B) The mean absolute $IC_{50}$ value for MV4;11 cells and 95% confidence interval [n = 5] for this replication attempt is plotted with the $IC_{50}$ value reported in *Dawson et al. (2011)* displayed as a single point (red circle) for comparison. (C) Dose response curves for each biological repeat [n = 4] for the K-562 cell line. Percent cell viability is relative to DMSO treated cells with I-BET151 dose range from 1 pM to 1 mM. The $IC_{50}$ estimates were not able to be accurately determined following published guidelines (*Sebaugh, 2011*). Additional details for this experiment can be found at https://osf.io/zm3j4/.

The following figure supplements are available for figure 1:

**Figure supplement 1.** Cellular I-BET151 dose response curves for each MV4;11 biological repeat.

**Figure supplement 2.** Cellular I-BET151 dose response curves for each K-562 biological repeat.

The analysis plan specified in the Registered Report (*Fung et al., 2015*) proposed to compare the $IC_{50}$ values from MV4;11 cells to those determined for K-562 cells. However, as stated above this could not be performed because of the inability to determine the K-562 $IC_{50}$ values. Instead, we performed an exploratory analysis using the MV4;11 $IC_{50}$ values compared to a constant of 100 µM, the highest dose in K-562 cells before inhibition started to be observed, which was statistically significant (one-sample $t$-test; $t(4)$ = 20.63, $p$=3.26×$10^{-5}$; Cohen's $d$ = 9.23, 95% CI [3.13, 15.45]). To summarize, for this experiment we found results that were in the same direction as the original study and statistically significant where predicted.

## Analysis of *BCL2* gene expression following I-BET151 treatment

A key antiapoptotic gene, *BCL2*, which has been implicated in MLL-fusion leukaemias (*Robinson et al., 2008*; *Wang et al., 2011*), was reported in *Dawson et al. (2011)* to be selectively inhibited by I-BET151, suggesting a possible mode of action. We sought to replicate this experiment, which is similar to what was reported in Figure 3D of *Dawson et al. (2011)* and described in Protocol 3 in the Registered Report (*Fung et al., 2015*). While the original study included multiple leukaemia cell lines with an MLL-fusion, this replication attempt was restricted to the MV4;11 cell line, which was utilized in other experiments. Similar to the cellular $IC_{50}$ experiment, the K-562 cell line served as a control for the specificity towards MLL-fusion leukaemias. We used quantitative real-time-polymerase chain reaction (qRT-PCR) to assess *BCL2* expression in MV4;11 and K-562 cells treated with I-BET151 or vehicle control (*Figure 2*). Using the $2^{-\Delta\Delta Ct}$ method, treatment of MV4;11 cells with I-BET151 resulted in a 0.501 [n = 3, $SD$ = 0.048] mean fold change in *BCL2* expression relative to vehicle control, while K-562 cells remained largely unchanged [n = 3, M = 1.06, $SD$ = 0.028]. This is similar to *Dawson et al. (2011)*, which reported an ~0.22 mean fold change in *BCL2* expression for MV4;11 cells and an ~0.94 mean fold change for the K-562 cell line.

To provide a direct comparison to the original data, we are reporting the analysis specified *a priori* in the Registered Report (*Fung et al., 2015*). We planned to conduct one two-sample $t$-test, between K-562 and MV4;11 fold gene expression values, and two one-sample $t$-tests, on K-562, or MV4;11, fold gene expression values compared to a constant of 1, which represents the DMSO treated gene expression. To account for these multiple comparisons, the Bonferroni correction was used, making the *a priori* Bonferroni adjusted significance threshold 0.0167. We performed an unpaired, two-sample $t$-test on the fold gene expression values from K-562 cells compared to MV4;11 cells, which was statistically significant ($t(4)$ = 17.23, uncorrected $p$=6.66×$10^{-5}$, corrected $p$=0.0004). Thus, the null hypothesis that I-BET151 treatment on *BCL2* expression is similar for MV4;11 and K-562 cells can be rejected. Additionally, we performed a one-sample $t$-test on the fold gene expression values from K-562 cells compared to a constant of 1 ($t(2)$ = 3.53, uncorrected $p$=0.072, corrected $p$=0.216), and a one-sample $t$-test on the fold gene expression values from MV4;11 cells compared to a constant of 1 ($t(2)$ = 17.86, uncorrected $p$=0.003, corrected $p$=0.009). To summarize, for this experiment we found results that were statistically significant where predicted and in the same direction as the original study.

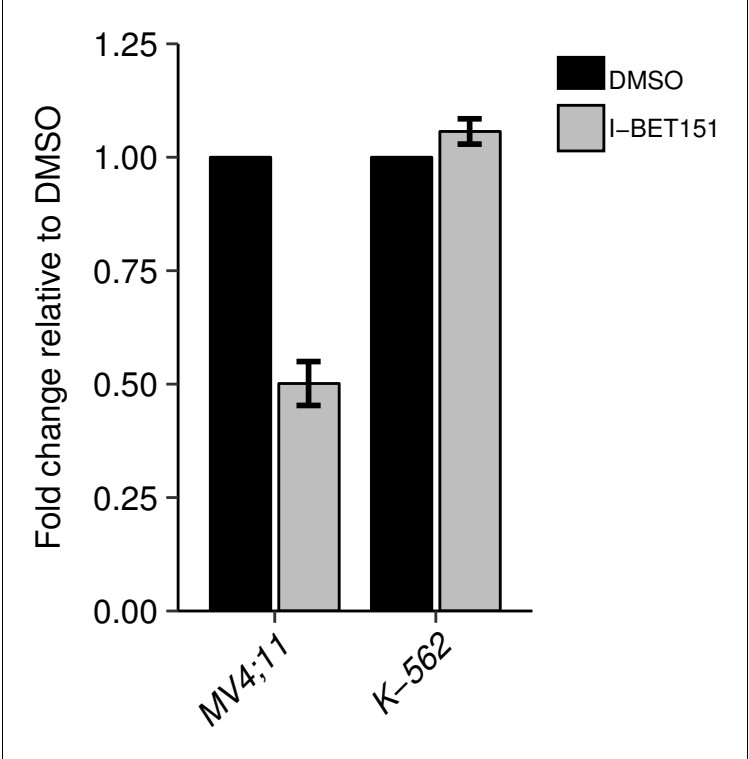

**Figure 2.** *BCL2* expression in I-BET151 treated MV4;11 and K-562 cells. MV4;11 and K-562 cells were treated with 500 nM I-BET151, or an equivalent volume of DMSO. Total RNA was isolated 6 hr after treatment and qRT-PCR was used to detect *BCL2* and *B2M* expression. Fold change in *BCL2* expression normalized to *B2M* and relative to DMSO is presented for I-BET151 treated MV4;11 and K-562 cells. Expression level of *BCL2* in DMSO was assigned a value of 1. Means reported and error bars represent *SD* from three independent biological repeats. Two-sample *t*-test comparing fold gene expression values from MV4;11 cells to K-562 cells; $t(4) = 17.23$, uncorrected $p=6.66\times10^{-5}$, *a priori* Bonferroni adjusted significance threshold = 0.0167; (Bonferroni corrected $p=0.0004$). One-sample *t*-test comparing fold gene expression from K-562 cells to a constant of 1 (DMSO treated cells); $t(2) = 3.53$, uncorrected $p=0.0719$, *a priori* Bonferroni adjusted significance threshold = 0.0167; (Bonferroni corrected $p=0.216$). One-sample *t*-test comparing fold gene expression from MV4;11 cells to a constant of 1 (DMSO treated cells); $t(2) = 17.86$, uncorrected $p=0.003$, *a priori* Bonferroni adjusted significance threshold = 0.0167; (Bonferroni corrected $p=0.009$). Additional details for this experiment can be found at https://osf.io/np6gq/.

## Generation of disseminated xenograft MLL mouse model and testing of I-BET151 compound *in vivo*

In order to test the efficacy of I-BET151 as a therapeutic agent in MLL-fusion leukaemia, we sought to replicate an experiment similar to what was reported in Figure 4B and D and Supplementary Figure 16A (*Dawson et al., 2011*) and described in Protocols 4–5 in the Registered Report (*Fung et al., 2015*). Prior to conducting this efficacy experiment, we determined the maximum tolerated dose (MTD) of I-BET151 in this xenograft mouse model, which imitates the pathology of human leukaemia by disseminating to the bone marrow (*Lopes de Menezes et al., 2005*). Five weeks after intravenous injection of MV4;11 cells into 7-week-old female NOD-SCID mice, the animals were successfully engrafted as determined by detectable human leukaemia cells in the peripheral blood. This was an increase from the 3 weeks reported in the original study, but was necessary due to the lack of detectable leukaemia cells in mice at 3 weeks post-injection during this replication attempt. Mice were randomized into treatment groups and after 21 days of daily intraperitoneal (IP) injections of vehicle control or three different doses of I-BET151, the MTD was determined as 20 mg/kg/day (*Figure 3*) based on the criteria pre-specified in the Registered Report (*Fung et al., 2015*). For comparison the original study delivered I-BET151 daily at 30 mg/kg. While it is not uncommon for variations in stock solution to cause differing compound potency and toxicity (*Kannt and Wieland, 2016*), a

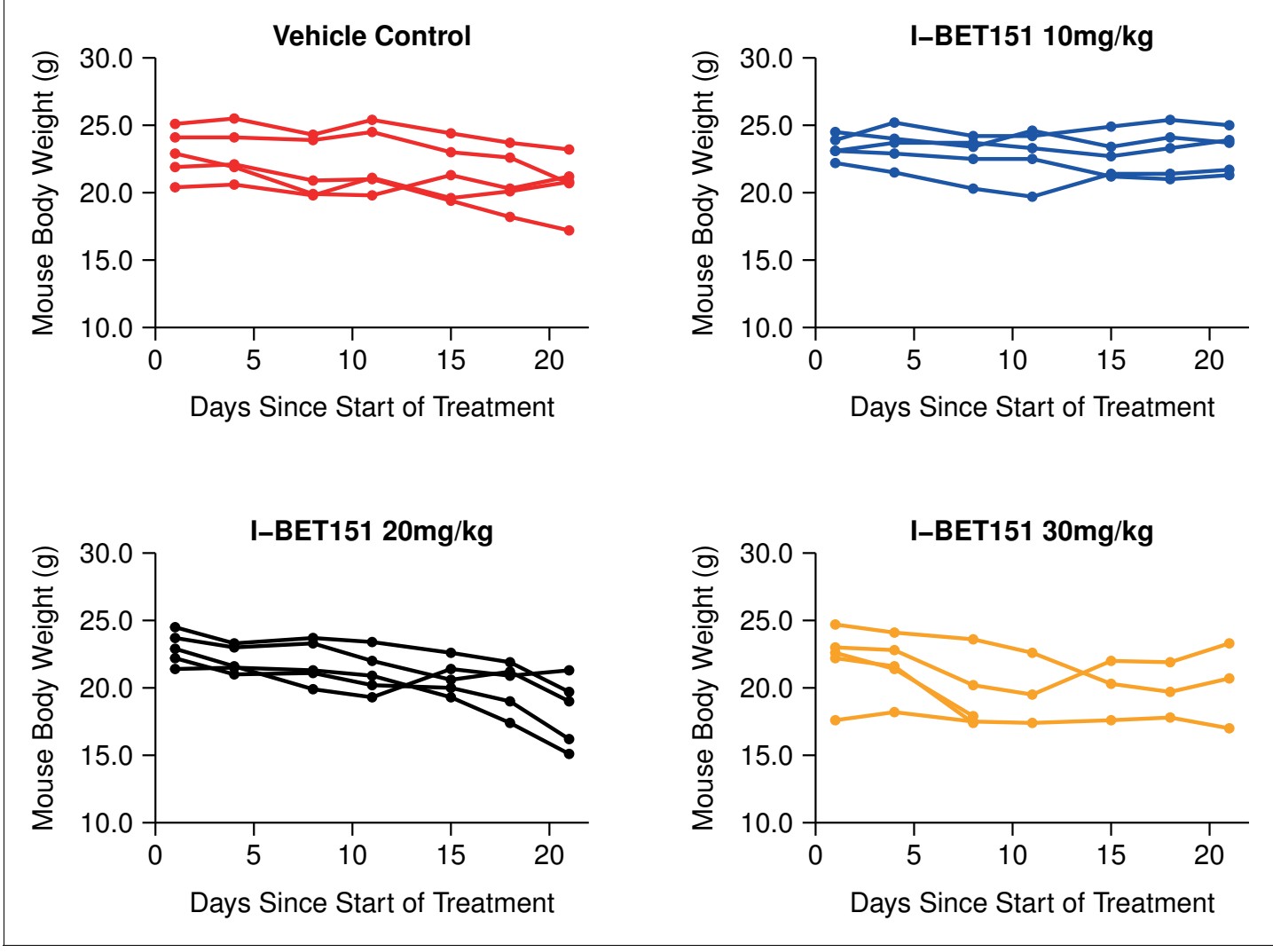

**Figure 3.** Maximal tolerated dose in an I-BET151 treated xenograft mouse model of MLL-fusion leukaemia. Female NOD-SCID mice were conditioned with Busulfan and intravenously injected with $1 \times 10^7$ MV4;11 cells in vehicle (PBS). Once disease was established (detectable MV4;11 cells from retro-orbital bleeds), mice were randomized into four cohorts and received daily IP injections of vehicle control [n = 5], 10 mg/kg I-BET151 [n = 5], 20 mg/kg I-BET151 [n = 5], or 30 mg/kg I-BET151 [n = 5]. The animals were monitored and their weight was measured for the duration of the 21 day treatment. The Y axis represents mouse weight in grams, the X axis represents days since the start of treatment. Additional information can be found at https://osf.io/juzmg/.

The following figure supplement is available for figure 3:

**Figure supplement 1.** Maximal tolerated dose in an I-BET151 treated xenograft mouse model of MLL-fusion leukaemia.

reduced dose could have an impact on I-BET151 efficacy. Indeed, recent work from authors of the original study reported I-BET151 administered at 10 mg/kg had no impact on survival in a MLL-AF9 leukaemia mouse model (*Gilan et al., 2016*), while I-BET151 administered at 15 mg/kg did provide a survival benefit in a mouse model of NPM1c AML, albeit with a smaller effect size (*Dawson et al., 2014*).

Following the same procedure as the MTD protocol, another cohort of 7-week-old female NOD-SCID mice were injected with MV4;11 cells and monitored until engraftment was established, which also occurred 5 weeks after injection. Mice were treated daily with I-BET151 at 20 mg/kg or vehicle control by IP injection for up to 21 days or until mice were humanely euthanized at signs of distress or disease (e.g. hind limb paralysis). At the time of sacrifice, disease presence was determined by

evaluating the percentage of human leukaemia cells present in the peripheral blood, spleen, and bone marrow (*Figure 4B*). Disease was defined, *a priori*, as 0.5% of human leukaemia cells over the total live nucleated cells, which included human and mouse cells, in the sample. After the start of treatment, mice treated with vehicle control achieved a median survival of 19.0 days [n = 11], which is indistinguishable from the 18.5 days [n = 12] observed in mice treated with I-BET151 (*Figure 4A*). A planned comparison between these survival distributions using a log-rank (Mantel-Cox) test was not statistically significant (*p*=0.536). This compares to the original study that reported a median survival of 14 days, [n = 5] for vehicle control treated animals that was lengthened to the experimental end-point of 21 days with I-BET151 treatment. While the median survival after the start of treatment in control treated mice are similar between these studies, the overall median survival from the MV4;11 injection are farther apart, with the original study reporting a median time of 35 days and this replication attempt reporting 59 days due to the added time from injection to beginning of treatment. Comparatively, other published studies using this model have reported a range of median survival times, such as 41 days (*O'Farrell et al., 2003*), 51 days (*Dorn et al., 2009*; *Lopes de Menezes et al., 2005*; *Ma et al., 2015*), and 60 days (*Hardwicke et al., 2009*) after intravenous injection of MV4;11 cells into NOD-SCID mice. To summarize, for this experiment we found the survival results were not in the same direction as the original study and not statistically significant where predicted.

In addition to monitoring survival, disease burden was determined for vehicle control and I-BET151 treated mice. For the efficacy study, the median disease burden observed in the peripheral blood, bone marrow, and spleen were smaller in I-BET151 treated mice compared to the control mice (*Figure 4B,C*). This was also observed in the MTD study (*Figure 3—figure supplement 1*). For example, the median percent of leukaemia cells detected in the bone marrow of vehicle control mice was 10.7%, which was reduced to 5.3% in I-BET151 treated mice (*Figure 4B*). This compares to the representative example reported in Supplementary Figure 16A of *Dawson et al. (2011)*, which reported 14% leukaemia burden in the bone marrow of the control mouse and 1.5% in the I-BET151 treated mouse where low-level residual leukaemia was reported after the 21 day treatment. To summarize, for the disease burden aspect of this experiment we found the results were in the same direction as the original study.

To assess if I-BET151 impacted cell death, specifically apoptosis and necrosis, flow cytometric analysis was also performed on the isolated cells from the vehicle control or I-BET151 treated mice. The degree of apoptosis and necrosis was assessed within the leukaemia population by staining with Annexin V and propidium iodide (PI). Observationally, there was no notable difference in the peripheral blood or spleen, with a minor decrease in both apoptotic (Annexin V+PI-) and necrotic cells (Annexin V+PI+) observed in the bone marrow of I-BET151 treated mice compared to vehicle control (*Figure 4—figure supplement 1*).

Unexpectedly, at the time of sacrifice, animal health issues were observed beyond the anticipated disease presentation for this model. These included gastrointestinal bloating in three of the I-BET151 treated mice and solid tumors detected in four of the vehicle control treated mice and two of the I-BET151 treated mice. While it has been previously noted that NOD-SCID mice have a high rate of spontaneous thymic lymphomas, no other tumor types were previously reported and tumors did not manifest until 20 weeks of age (*Prochazka et al., 1992*). To explore the origin of the observed tumors, sections were stained using a human leukocyte antigen (HLA-A,B,C) antibody (*Figure 4—figure supplement 2*). In all tested samples there was extensive staining, indicating the injected human leukaemia cells spread throughout the mouse outside the hematological tissues. This suggests these mice most likely succumbed to fulminant leukaemia manifesting often as extramedullary disease (EMD), a diagnosis that has been observed in a subset of patients with various types of AML (*Ganzel et al., 2016*). Indeed, the original study reported EMD in the majority of the control mice, in contrast to none of the I-BET151 treated mice (*Dawson et al., 2011*). Collectively, these observations raise the possibility that although a survival benefit was not observed with I-BET151 treatment in this replication attempt, despite an overall reduced disease burden, that health issues not directly related to leukaemia could have impacted mortality observed in some of the animals, particularly those treated with I-BET151. Additionally, busulfan conditioning could have influenced engraftment kinetics and disease progression in this animal model and potentially impacted the responsiveness of MV4;11 cells to I-BET151 treatment.

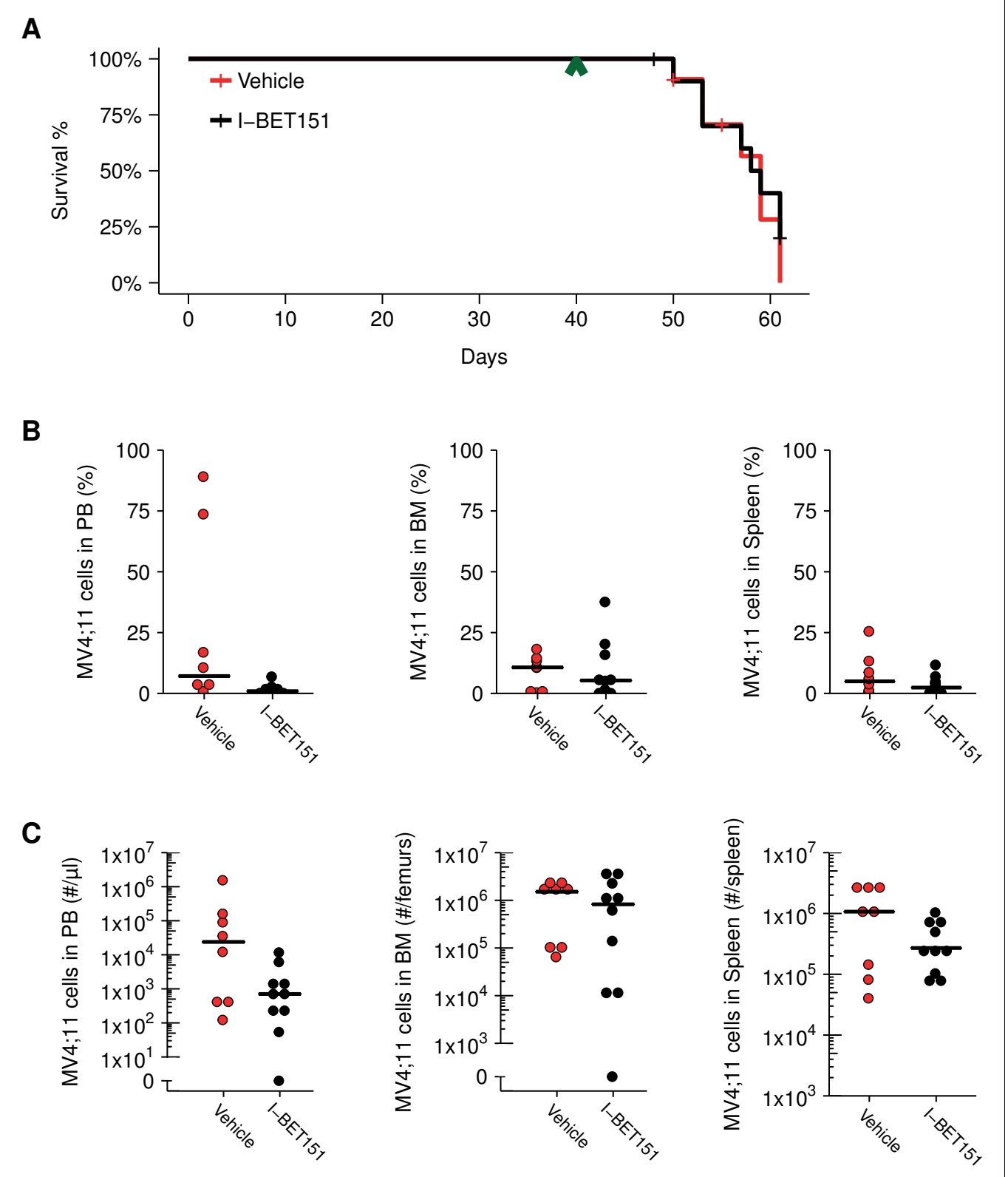

**Figure 4.** Efficacy study of I-BET151 in xenograft mouse model of MLL-fusion leukaemia. Female NOD-SCID mice were xenotransplanted with $1 \times 10^7$ MV4;11 cells after conditioning with Busulfan. Following establishment of disease (detectable MV4;11 cells from retro-orbital bleeds), mice were randomly assigned to receive daily IP injections of 20 mg/kg I-BET151 or vehicle control. (**A**) Kaplan-Meier plot of survival during the course of the study. Green arrowhead indicates when treatment commenced on day 40 with treatment continuing for the pre-specified period of 21 days. Animals

*Figure 4 continued on next page*

*Figure 4 continued*

with no detectable disease at the time of sacrifice (less than 0.5% of MV4;11 cells) or that were unable to be evaluated at the time of death were censored (denoted by a cross). Number of mice monitored: n = 11 for vehicle group and n = 12 for I-BET151 group. Log-rank (Mantel-Cox) test of I-BET151 treatment compared to vehicle control (*p*=0.536). (**B–C**) At the time of sacrifice, disease burden was evaluated in peripheral blood (PB), bone marrow (BM), and spleen cells. Number of mice analyzed: n = 8 for vehicle group and n = 10 for I-BET151 group. (**B**) The percent of MV4;11 cells was determined by flow cytometric analysis as the percent of HLA-A,B,C$^+$ cells in the total nucleated population (both mouse and human cells). Dot plot with medians reported as crossbars. (**C**) The absolute number of MV4;11 cells each sample was also determined using CountBright absolute counting beads. Dot plot with medians reported as crossbars. Additional details for this experiment can be found at https://osf.io/jakpw/.

The following figure supplements are available for figure 4:

**Figure supplement 1.** Mouse body weight and cell death analysis in I-BET151 treated xenograft mouse model of MLL-fusion leukaemia.
**Figure supplement 2.** Detection of HLA by immunohistochemistry in mouse tissues.
**Figure supplement 3.** Flow cytometry gating strategies.

## Meta-analyses of original and replicated effects

We performed a meta-analysis using a random-effects model, where possible, to combine each of the effects described above as pre-specified in the confirmatory analysis plan (*Fung et al., 2015*). To provide a standardized measure of the effect, a common effect size was calculated for each effect from the original and replication studies. Cohen's *d* is the standardized difference between two independent means using the pooled sample standard deviation. For a one-sample test, Cohen's *d* is the difference between the sample mean and the null value using the sample standard deviation. The hazard ratio (HR) is the ratio of the probability of a particular event, in this case death, in one group compared to the probability in another group. The estimate of the effect size of one study, as well as the associated uncertainty (i.e. confidence interval), compared to the effect size of the other study provides another approach to compare the original and replication results (*Errington et al., 2014*; *Valentine et al., 2011*). Importantly, the width of the confidence interval for each study is a reflection of not only the confidence level (e.g. 95%), but also variability of the sample (e.g. *SD*) and sample size.

A meta-analysis of the I-BET151 IC$_{50}$ values in MV4;11 and K-562 cells was not conducted since the original study reported single IC$_{50}$ values. Comparing the original and replication results, the IC$_{50}$ value reported in *Dawson et al. (2011)* for MV4;11 cells fell outside the 95% CI of the values generated during this replication attempt (*Figure 1B*). In K-562 cells, both the original and replication studies could not accurately determine the IC$_{50}$ estimates indicating K-562 cells are largely unaffected by I-BET151.

There were three comparisons made with the qRT-PCR data analyzing *BCL2* expression following I-BET151 treatment (*Figure 5A,B*). For two of the comparisons, the fold *BCL2* expression in each cell line was compared to a constant of 1, which represents the DMSO treated group (*Figure 5A*). The comparison of K-562 fold gene expression compared to a constant of 1 resulted in *d* = −2.04, 95% CI [−4.17, 0.13] in this study, whereas *d* = 1.30, 95% CI [−0.38, 2.88] for the data estimated *a priori* from *Figure 3D* in the original study (*Dawson et al., 2011*). A meta-analysis of these effects resulted in *d* = −0.307, 95% CI [−3.57,2.96], which was not statistically significant (*p*=0.854). The results are in opposition when considering the direction of the effect. The point estimate of the replication effect size was not within the confidence interval of the original result, nor was the point estimate of the original effect size within the confidence interval of the replication result. The comparison of MV4;11 fold gene expression compared to a constant of 1, resulted in *d* = 10.31, 95% CI [1.54, 19.86] in this replication attempt, *d* = 26.00, 95% CI [4.10, 48.25] for the data estimated from the original study (*Dawson et al., 2011*) and *d* = 15.13, 95% CI [0.94, 29.32] for a meta-analysis of the two effects. Both the original and replication results are consistent when considering the direction of the effect, with the point estimate of the replication effect size was within the confidence interval of the original result and vice versa. Further, the random effects meta-analysis resulted in a statistically significant effect (*p*=0.037). For the third comparison between MV4;11 and K-562 fold gene expression, this replication attempted resulted in *d* = 14.07, 95% CI [4.73, 23.59], which compares to

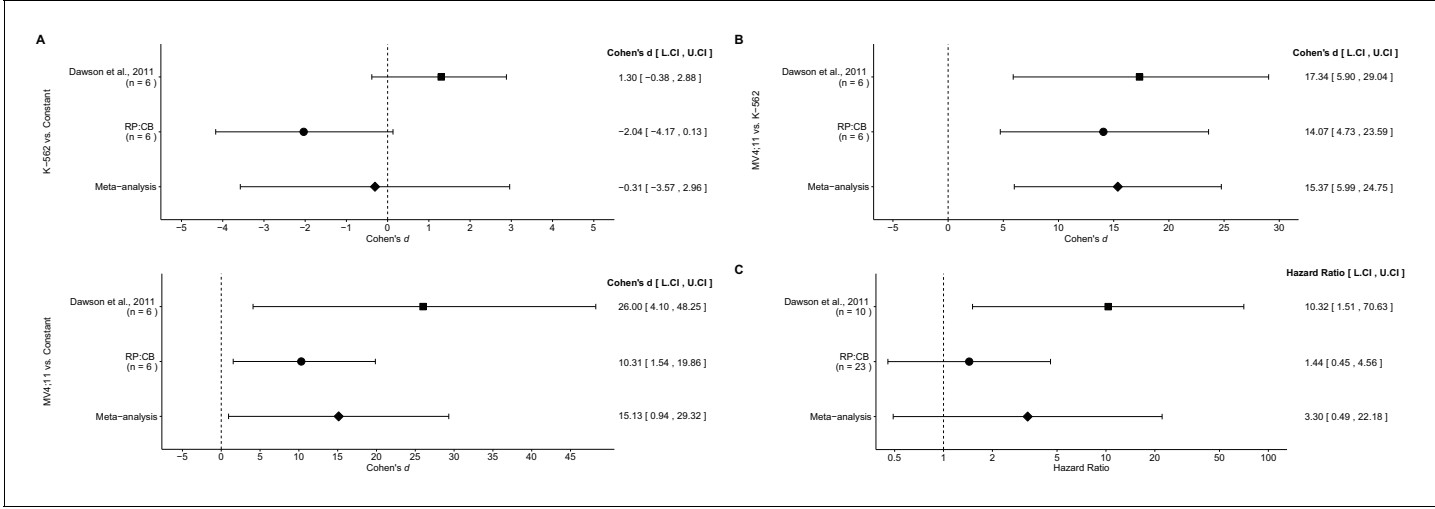

**Figure 5.** Meta-analyses of each effect. Effect size and 95% confidence interval are presented for *Dawson et al. (2011)*, this replication study (RP:CB), and a random effects meta-analysis of those two effects. Sample sizes used in *Dawson et al. (2011)* and this replication attempt are reported under the study name. (**A**) Fold *BCL2* expression in K-562 cells compared to a constant of 1 (DMSO) (meta-analysis p=0.854), and fold *BCL2* expression in MV4;11 cells compared to a constant of 1 (DMSO) (meta-analysis p=0.037). (**B**) Fold *BCL2* expression in MV4;11 cells compared to the fold *BCL2* expression in K-562 cells (meta-analysis p=0.001). (**C**) HR for mice treated daily with I-BET151 compared to vehicle control (meta-analysis p=0.220). Additional details for these meta-analyses can be found at https://osf.io/vfp47/.

$d$ = 17.34, 95% CI [5.90, 29.04] for the data estimated from the original study (*Dawson et al., 2011*) (*Figure 5B*). A meta-analysis of these effects resulted in $d$ = 15.37, 95% CI [5.99, 24.75], which was statistically significant (*p*=0.001). Additionally, both results are consistent when considering the direction of the effect with both effect size point estimates falling within the confidence interval of the other study.

The comparison of the overall survival distributions between mice treated with I-BET151 compared to those that were treated with vehicle control resulted in a HR of 1.44, 95% CI [0.45, 4.56] for this replication attempt compared to a HR of 10.32, 95% CI [1.51, 70.63] for the original study (*Dawson et al., 2011*). A meta-analysis (*Figure 5C*) of these effects resulted in a HR of 3.30, 95% CI [0.49,22.18], which was not statistically significant (*p*=0.220). Both results are consistent when considering the direction of the effect, however the point estimate of the replication effect size was not within the confidence interval of the original result, and vice versa. The large confidence intervals of the meta-analysis along with a statistically significant Cochran's *Q* test (*p*=0.085), suggests heterogeneity between the original and replication studies.

This direct replication provides an opportunity to understand the present evidence of these effects. Any known differences, including reagents and protocol differences, were identified prior to conducting the experimental work and described in the Registered Report (*Fung et al., 2015*). However, this is limited to what was obtainable from the original paper and through communication with the original authors, which means there might be particular features of the original experimental protocol that could be critical, but unidentified. So while some aspects, such as cell lines, strain of mice, number of cells plated/injected, and specific BET inhibitor were maintained, others were changed at the time of study design (*Fung et al., 2015*) or in the execution of the replication. This includes differences such as the conditioning for xenotransplantation (sub-lethal irradiation to busulfan) and dose of I-BET151 used for the efficacy study (30 mg/kg to 20 mg/kg). The regimen used in this replication attempt has not been fully evaluated with MV4;11 cells regarding its impact on disease progression and treatment responsiveness, an important consideration, since the method of conditioning can influence overall and hematological toxicity, as well as overall survival, despite similar engraftment levels (*Robert-Richard et al., 2006*; *Saland et al., 2015*). Furthermore, other aspects were unknown or not easily controlled for. These include variables such as cell line genetic drift (*Hughes et al., 2007*; *Kleensang et al., 2016*), mouse sex (*Clayton and Collins, 2014*), circadian biological responses to therapy (*Fu and Kettner, 2013*), the microbiome of recipient mice

(*Macpherson and McCoy, 2015*), housing temperature in mouse facilities (*Kokolus et al., 2013*), and differing compound potency resulting from different stock solutions (*Kannt and Wieland, 2016*) or from variation in cell division rates (*Hafner et al., 2016*). Whether these or other factors influence the outcomes of this study is open to hypothesizing and further investigation, which is facilitated by direct replications and transparent reporting.

## Materials and methods

As described in the Registered Report (*Fung et al., 2015*), we attempted a replication of the experiments reported in Figures 2A, 3D, 4B and D, and Supplementary Figures 11A-B and 16A of *Dawson et al. (2011)*. A detailed description of all protocols can be found in the Registered Report (*Fung et al., 2015*). Additional detailed experimental notes, data, and analysis are available on the Open Science Framework (OSF) (RRID:SCR_003238) (https://osf.io/hcqqy/; *Shan et al., 2017*).

### Cell culture

MV4;11 (ATCC, cat# CRL-9591, RRID:CVCL_0064) and K-562 cells (ATCC, cat#CCL-243, RRID:CVCL_0004) were maintained in RPMI-1640 medium supplemented with 10% fetal bovine serum (FBS) and 1% penicillin/streptomycin at 37°C in a humidified atmosphere at 5% $CO_2$. Quality control data for both the MV4;11 and K-562 cell lines related to the *in vitro* experiments are available at https://osf.io/vb9q4/ and quality control data for the MV4;11 cell line related to the *in vivo* experiments are available at https://osf.io/5tvwe/. This includes results confirming the cell lines were free of mycoplasma contamination and common mouse pathogens. Additionally, STR DNA profiling of the cell lines was performed and all cells were confirmed to be the indicated cell lines when queried against STR profile databases.

### Therapeutic compounds

25 mg of I-BET151 (Sigma-Aldrich, cat# SML0666; lot# 033M4620V) was dissolved in 601.8 µl filtered dimethyl sulfoxide (DMSO) to generate a 0.1 M stock, which was stored at 4°C until use for *in vitro* experiments. For *in vivo* experiments, I-BET151 (lot# 084M4722V) was dissolved in DMSO to generate a 60 mg/ml stock. This was further diluted to the appropriate working solution with vehicle (10% wt/vol Kleptose HPB in 0.9 %/g NaCl, pH 5.0), filter sterilized, and aliquoted (21 total) into glass vials. Final concentration of DMSO in the working solution was 5% (v/v). Aliquots were stored at 4°C until use when aliquots were brought to room temperature before injection.

### Cellular dose response assay

The doubling time of MV4;11 and K-562 cells were empirically determined by seeding at $4 \times 10^4$ cells per well into two 96-well plates. The day after seeding and another three days later an MTS cell proliferation assay (Promega CellTiter-Aqueous One, cat# G3582) was performed according to the manufacturer's instructions with a 4 hr incubation at 37°C. Absorbance was read at 490 nm and the plate background was subtracted at 650 nm using a plate reader (Molecular Devices, SpectraMax 190) and Softmax Pro data acquisition and analysis software (RRID:SCR_014240), version 3.1.2. The doubling time of each cell line was calculated to be 48 hr and 36.2 hr for the MV4;11 and K-562 cell lines, respectively. Doubling time was calculated using the formula (doubling time = incubation time*ln(2)/ln(second plate average reading/first plate average reading)).

For cellular dose response, MV4;11 and K-562 cells were seeded at $4 \times 10^4$ cells per well (non-edge wells) into a 96-well plate and incubated overnight. Cells were treated with serial dilutions of I-BET151 to yield 10 dilutions ranging from 0.001 nM to 1 mM, which differs from the conditions listed in the Registered Report due to solubility limits. I-BET151 doses from 0.001 nM to 100 µM were diluted with medium to give a final DMSO concentration of 0.1%, while the 1 mM concentration gave a final DMSO concentration of 1% due to drug solubility. Both concentrations of DMSO were used to treat control cells in addition to medium alone wells that were used for background subtraction. All conditions were done in technical triplicate. Following an incubation of 3 population doublings of the cell lines, an MTS cell proliferation assay was performed as described for the doubling time experiment. Percent viability was calculated as a percentage of DMSO treated cells after background subtraction (medium only wells). Values for each biological repeat were fit to a four-

parameter logistic curve and absolute $IC_{50}$ values were calculated with R software (RRID:SCR_001905), version 3.3.2 (*R Core Team, 2016*).

## Gene expression analysis

MV4;11 and K-562 cells were seeded at $1 \times 10^5$ cells per well in a 48 well plate in triplicate for each condition and incubated overnight. Cells were then treated with either 500 nM I-BET151 or an equivalent volume of DMSO (final concentration of 0.1% (v/v)) for 6 hr. Cells were then harvested and RNA isolated using the RNAspin mini kit (GE Healthcare, cat# 25-0500-70) according to manufacturer's instructions. RNA concentration and purity was determined (data available at https://osf.io/n8frz/) and samples were stored at −80°C until further analyzed. Total RNA was reverse transcribed into cDNA using SuperScript III First-Strand Synthesis System (GE Healthcare, cat# 27-9261-01) according to manufacturer's instructions. qRT-PCR reactions were then performed in technical triplicate using *BCL2* and *B2M*-specific primers (sequences listed in Registered Report [*Fung et al., 2015*]), and SYBR Green PCR Master Mix (ThermoFisher, cat# 4344463) according to manufacturer's instructions. Reaction volume was 20 µl and consisted of 10 µl 2X SYBR Green PCR Master Mix, 20 µM forward and reverse primers, and 5 µl RNA diluted in water. A negative control without RNA template was also included. PCR cycling conditions were used as follows: [1 cycle 95°C for 10 min – 40 cycles 95°C for 15 s, 60°C 60 s] using a real-time PCR system (Bio-Rad, DNA Engine Opticon System) and Opticon Monitor Software (Bio-Rad, RRID:SCR_014241), version 1.4. *BCL2* transcript levels were normalized to *B2M* levels in each sample and I-BET151 treated samples were compared to DMSO treated samples for each cell line using the ΔΔCt method.

## Animals

All animal procedures were approved by the University of Pennsylvania Stem Cell and Xenograft Core IACUC# 803506 and were in accordance with the University of Pennsylvania policies on the care, welfare, and treatment of laboratory animals. Four-week old female NOD-SCID mice (Jackson Laboratory, Stock No: 001303, RRID:IMSR_JAX:001303) were housed in sterile conditions using high-efficiency particulate arrestance (HEPA)-filtered micro-isolator with 12 hr light/dark cycles, and fed with sterile rodent chow and acidified water *ad libitum*. The mice were housed for approximately 3 weeks before being enrolled in the study.

## I-BET151 maximum tolerated dose (MTD) study

After acclimation, 20 female NOD-SCID mice were conditioned with 30 mg/kg Busulfan (Otsuka America Pharmaceutical, Inc., cat# NDC 59148-070-90) by intraperitoneal (IP) injection 24 hr prior to injection of MV4;11 cells. Conditioned mice were then intravenously injected via the tail vein with $1 \times 10^7$ MV4;11 cells in 0.2 ml sterile vehicle (PBS) (Gibco, Life Technologies, cat# 14190–136). Mice were inspected daily for signs of distress and the 'Mouse Health Scoring System' (*Cooke et al., 1996*) was used to record scores for each mouse over the course of the study. At day 21 post-injection, retro-orbital bleeds were collected and analyzed for leukaemia burden by flow cytometry. Detectable disease was not achieved on day 21 and daily inspection continued until day 34 when disease was detected. Animals were then ranked according to disease burden (percent HLA-A,B,C$^+$ cells) and randomly assigned to one of four groups using an alternating serpentine method. Group assignment was done by simple randomization. All mice had detectable leukaemia at time of randomization and variability of disease burden was evenly distributed among the four conditions (One-way ANOVA; $F(3,16) = 0.019$, $p=0.996$). After randomization, mice received daily IP injections of vehicle control, 10 mg/kg I-BET151, 20 mg/kg I-BET151, or 30 mg/kg I-BET151 in a volume of 10 ml/kg body weight. Injection volume and dosing was kept constant throughout the treatment period. Mice were inspected daily for the duration of the 21 day drug treatment for decreased spontaneous activity (grooming and ambulation), ruffled fur, weight loss, and loss of hind limb motility. Mice were sacrificed if moribund, received a health-monitoring score of three, or within three days of the last treatment. Upon sacrifice, cardiac puncture was performed to collect the peripheral blood, and then the spleen and both tibias and femora were removed. Splenic cell suspension was made in cold PBS by gently mushing the spleen between two microscopic glass slides. Bones were flushed with cold PBS to isolate bone marrow cells. Red blood cells from samples were lysed using

ammonium chloride solution (STEMCELL Technologies, cat# 07850) according to manufacturer's instructions prior to flow cytometry analysis.

## I-BET151 efficacy study

After one week of acclimation, 28 female NOD-SCID mice were conditioned with Busulfan and injected with MV4;11 cells as described in the MTD study. Five mice died unexpectedly before the first retro-orbital bleed to assess leukaemia burden. The remaining 23 mice were analyzed on day 21 post-injection, with no engraftment detected. Similar to the MTD study, mice were analyzed again two weeks later, with leukaemia burden detected. Animals were then ranked according to disease burden (percent HLA-A,B,C$^+$ cells) and randomly assigned to a group using an alternating serpentine method. Designation of vehicle control or I-BET151 to group was done by a simple randomization. All mice had detectable leukaemia at time of randomization and variability of disease burden was evenly distributed among the two conditions (Student's $t$-test; $t(21)$ = 0.028, $p$=0.978). After randomization, mice received daily IP injections of vehicle control or 20 mg/kg I-BET151 in a volume of 10 ml/kg body weight. Injection volume and dosing was kept constant throughout the treatment period. Mice were inspected daily and peripheral blood, spleen, and bone marrow cells were isolated upon sacrifice as described for the MTD study.

## Flow cytometry analysis

To assess leukaemia burden in peripheral blood, spleen, and bone marrow cells, a maximum of 10$^6$ cells were stained in a sample volume of 50 µl each. 20 µl of APC conjugated anti-human HLA-A,B,C (Biolegend, cat# 311410, RRID:AB_314879) or APC conjugated isotype control (Biolegend, cat# 400220, RRID:AB_326468) was added and incubated at room temperature for 15 min. CountBright absolute counting beads (Life Technology, cat# C36950) in 1X FACS lysing solution were added and incubated at room temperature for an additional 15 min. Flow cytometry analysis was performed on a Canto or LSRII (Becton-Dickinson) and analyzed with FlowJo software (Tree Star, Inc., RRID:SCR_008520), version 10.1. Gating strategy was described in the Registered Report with a representative example reported in *Figure 4—figure supplement 3A*.

To assess cell death, a maximum of 10$^6$ cells were stained in a sample volume of 50 µl each. 5 µl of APC conjugated anti-human HLA-A,B,C or APC conjugated isotype control was added and incubated at 4°C for 30 min. After washing twice with 1 ml of 1X Binding Buffer and collecting cells by spinning at 300x$g$ for 10 min, cells were resuspended in 100 µl of 1X Binding Buffer. 10 µl of Annexin V-FITC (Miltenyi Biotech Ltd, cat# 130-092-052) was added and incubated at room temperature for 15 min in the dark. After washing twice with 1 ml of 1X Binding Buffer and collecting cells by spinning at 300x$g$ for 10 min, cells were resuspended in 500 µl of 1X Binding Buffer. Prior to flow cytometry analysis 5 µl of propidium iodide (PI) (Invitrogen, cat# P3566) was added. An initial attempt with 7-AAD, which was pre-specified in the Registered Report, was unsuccessful because of an inability to properly compensate, thus PI was substituted. Gating strategy was described in the Registered Report with a representative example reported in *Figure 4—figure supplement 3B*. Flow cytometry data for this work is also available via Flow Repository (RRID:SCR_013779; *Spidlen et al., 2012*), where it is directly accessible at https://flowrepository.org/id/FR-FCM-ZY68.

## Immunohistochemistry

Tissues were fixed in 10% buffered formalin (Fisher Scientific, cat# 23–245685) and subsequently embedded in paraffin. The paraffin blocks were sectioned (5 µm) and stained with mouse anti-human Class 1 HLA-A,B,C antibody (Abcam, cat# ab70328, RRID:AB_1269092) and counterstained with hematoxylin. Staining was performed on a BOND-IIITM instrument (Leica Biosystems, Wetzlar, Germany) using the Bond Polymer Refine Detection System (Leica Biosystems, cat# DS9800). Heat-induced epitope retrieval was done for 20 min with ER1 solution (Leica Biosystems, cat# AR9961). Incubation with the HLA-A,B,C antibody was at 1:1200 dilution for 15 min followed by 8 min post-primary step and 8 min incubation with polymer HRP. Then endogenous peroxidase was blocked for 5 min followed by a 10 min incubation with DAB substrate. All staining steps were performed at room temperature with slides washed three times between each step with Bond Wash Solution (Leica Biosystems, cat# AR9590) or water. A section of human tonsil tissue was processed in parallel

as a positive control. Immunostaining was performed by the Pathology Clinical Service Center at the University of Pennsylvania, Perelman School of Medicine.

## Statistical analysis

Statistical analysis was performed with R software (RRID:SCR_001905), version 3.3.2 (*R Core Team, 2016*). All data, csv files, and analysis scripts are available on the OSF (https://osf.io/hcqqy/). Confirmatory statistical analysis was pre-registered (https://osf.io/kfx9p/) before the experimental work began as outlined in the Registered Report (*Fung et al., 2015*). Data were checked to ensure assumptions of statistical tests were met. A meta-analysis of a common original and replication effect size was performed with a random effects model and the metafor R package (*Viechtbauer, 2010*) (https://osf.io/vfp47/). The original study data was extracted *a priori* from the published figures by determining the mean and upper/lower error values for each data point. The extracted data were published in the Registered Report (*Fung et al., 2015*) and were used in the power calculations to determine the sample size for this study.

## Deviations from registered report

The range of I-BET151 doses used in cellular dose response assay was altered from 0.01 nM to 10 mM listed in the Registered Report to 1 pM to 1 mM reported in this replication attempt. This was due to the solubility of I-BET151 restricting the highest dose that could be tested in this replication attempt (1 mM), which was the same dose tested in the original study. For the *in vivo* experimentation, only female mice were used in this replication attempt rather than a male female split as outlined in the Registered Report. This deviation was caused by an inability to obtain male and female mice from the supplier when the MTD study was performed. To maintain consistency with the MTD study only female mice were utilized in the efficacy study. Furthermore, the time between MV4;11 injection and animal randomization was extended from the 21 days outlined in the Registered Report (*Fung et al., 2015*) due to undetectable disease burden at this time. Mice were checked for disease burden 2 weeks later to allow additional time for disease manifestation. This occurred in both the MTD and efficacy studies, with disease burden only becoming detectable after this additional time, after which the remaining aspects of the protocol were carried out.

For the cell death flow cytometry analysis an initial attempt using 7-AAD was unsuccessful because of an inability to properly compensate, thus PI, a suitable viability stain, was substituted. Additional materials and instrumentation not listed in the Registered Report, but needed during experimentation are also listed above.

## Acknowledgements

The Reproducibility Project: Cancer Biology would like to thank Dr. Mark Dawson (Peter MacCallum Cancer Centre) for sharing critical information and the following companies for generously donating reagents to the Reproducibility Project: Cancer Biology; American Type and Tissue Collection (ATCC), Applied Biological Materials, BioLegend, Charles River Laboratories, Corning Incorporated, DDC Medical, EMD Millipore, Harlan Laboratories, LI-COR Biosciences, Mirus Bio, Novus Biologicals, Sigma-Aldrich, and System Biosciences (SBI).

## Additional information

### Group author details

Reproducibility Project: Cancer Biology

Elizabeth Iorns: Science Exchange, Palo Alto, United States; Alexandria Denis: Center for Open Science, Charlottesville, United States; Stephen R Williams: Center for Open Science, Charlottesville, United States; Nicole Perfito: Science Exchange, Palo Alto, United States; Timothy M Errington, http://orcid.org/0000-0002-4959-5143: Center for Open Science, Charlottesville, United States

## Competing interests

XS, GD-D: Stem Cell and Xenograft Core, University of Pennsylvania, Perelman School of Medicine is a Science Exchange associated lab. JJF, AK: ProNovus Bioscience, LLC is a Science Exchange associated lab. RP:CB: EI, NP: Employed by and hold shares in Science Exchange Inc. The other authors declare that no competing interests exist.

## Funding

| Funder | Author |
| --- | --- |
| Laura and John Arnold Foundation | Reproducibility Project: Cancer Biology |

The funder had no role in study design, data collection and interpretation, or the decision to submit the work for publication.

## Author contributions

XS, JJF, AK, GD-D, Acquisition of data, Drafting or revising the article; RP:CB, Analysis and interpretation of data, Drafting or revising the article

## Ethics

Animal experimentation: All animal procedures were approved by the University of Pennsylvania Stem Cell and Xenograft Core IACUC# 803506 and were in accordance with the University of Pennsylvania policies on the care, welfare, and treatment of laboratory animals.

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
