## [Decision Letter]

Thank you for submitting your article "Replication Study: Inhibition of BET recruitment to chromatin as an effective treatment for MLL-fusion leukaemia" for consideration by *eLife*. Your article has been reviewed by three peer reviewers, and the evaluation has been overseen by a Reviewing Editor and Charles Sawyers as the Senior Editor. Two of the reviewers, Mark Dawson and M Dawn Teare, have agreed to share their names; a third reviewer remains anonymous.

The following issues need to be addressed before acceptance, as outlined below:

1) The Abstract and text needs to make clear that (and how) the in vivo protocol employed deviated from the original protocol and acknowledge how this might alter the outcome. It seems very important to not give the wrong impression that the xenograft experiments of the original study could not be reproduced, which would easily be the take-home-message after a "superficial read" of the current manuscript. To avoid this impression, the authors should clearly highlight the most relevant deviations from the original study (i.e. the alternative conditioning regimen and the reduced I-BET151 dose) in the Abstract.

2) Complications to interpretation due to other animal health issues should be clearly described. Here, it is not necessary to describe all potential issues in detail in the main text (although perhaps in supplement), but to address the following possibility, raised by reviewer #1. "The fact that the reduction in disease burden following IBET treatment closely paralleled the findings reported in the original study yet a survival benefit was not observed here raises the possibility that leukaemia related mortality was not the primary cause for the mortality observed in at least some animals in the IBET treatment arm."

Specific examples of places where clarification is in order are given below, as well as some additional suggestions, which have been extracted from the reviews:

Reviewer #1:

From a reader's perspective there is too much emphasis on the statistical testing performed and not enough discussion about the results and how variations to the original study should be interpreted.

For experiments 1) and 2) the general results obtained very much led to the same conclusion. In experiment 1) however, although there was, as previously, a marked difference in sensitivity between MV4;11 (IC50 9 nM) and K562 (unmeasurable), the meta-analysis demonstrated poor reproducibility between the concentrations in the original and reproduced experiments for MV4:11 (IC50 of 26 nM previously).

The main area of discrepancy relates to 3) the in vivo mouse experiment and specifically the efficacy study assessing overall survival. There are a number of methodological aspects of this study that differ substantially to the original study and these are not raised in the Abstract nor discussed adequately in the manuscript. Therapeutically, the dose of the drug was different (less, at 20 vs 30 mg/kg) as was the duration of therapy, with treatment starting at d35 instead of d21 as in the previous study. As disease activity was determined by examination of peripheral blood only and we know that this xenograft model gives disseminated disease, one interpretation could be that the results differed as a lower dose was used in the face of an increased disease bulk. Most importantly technically, the conditioning regime used was markedly different. 30mg/kg of Busulfan is a very different approach to the more conventional strategy of sub-lethal irradiation performed in the original study. The conditioning regime can dramatically alter engraftment kinetics and also the natural history of the disease. This is reflected in this particular case by the fact that one quarter of the mice on this study developed 'solid malignancies', which I assume to be extramedullary leukaemia. Furthermore 3/12 mice treated with I-BET developed gastrointestinal bloating and (whilst not explicitly described) presumably died for reasons that were independent of the leukaemia burden. This is alluded to by the authors when they state that "animal health issues were observed beyond the anticipated disease presentation of this model" but could be stressed more. These important issues impact significantly on an efficacy study that is essentially measuring all cause mortality. The fact that the reduction in disease burden following IBET treatment closely paralleled the findings reported in the original study yet a survival benefit was not observed here raises the possibility that leukaemia related mortality was not the primary cause for the mortality observed in at least some animals in the IBET treatment arm.

All of the above differences would also explain the apparently contradictory finding that, although survival was not impacted, disease bulk was consistently less in the IBET treated arm. A significant issue with the manuscript as currently presented is that the reader is left unaware of these critical details when reading the reproduction. It is essential to communicate this both in the Abstract, in the Results section and when discussing the discrepant results.

The results from the MTD study show that the mice in the 30mg/kg arm had the greatest variation in starting weight amongst all the cohorts. There is a 25% variation in weight between the heaviest and lightest mouse in this cohort whereas the starting weight of the mice in the other cohort's vary very little. The initial weight of the mice can contribute substantially to the observed toxicity of any intervention especially conditioning. This variation may be one of the reasons why more toxicity was observed in this cohort.

Reviewer #2:

The summary assessment of each experiment is not consistently written it states 'did not find a statistically significant' but the converse is not stated, i.e. 'we did see a statistically significant'. The summary needs to be consistent. This replication study has been powered from a statistical perspective so this needs to be clear in the Abstract.

Reviewer #3:

Most experiments reproduce key results of the original study, with the exception of in-vivo treatment studies in the MV4;11 AML xenograft model, where the authors only reproduce the impact on disease burden while impact on overall survival has been observed. For the interpretation of these results it should be noted that the replication study diverged from the original work in several experimental details that might be of relevance.

Overall, the following major points should be considered prior to publication:

In the replication of MV4;11 xenograft experiments several experimental conditions deviate from the original study. Most likely due to a lack of access to an irradiation unit, the authors conditioned NOD/SCID mice with a single dose of busulfan (30 mg/kg i.p. 24 hrs prior to MV4;11 transplantation), which has been reported to lead to comparable engraftment of human CD34+ cord blood cells. Apart from the difference in cell type, the original study by Robert-Richard et al. mainly used other dosing regimens (35 mg/kg 36 hrs prior to injection; or 2 x 25 mg 48/24 hrs prior to injection), which they found to be less toxic. While under these conditions, busulfan and irradiation performed similar in human CD34+ HSC engraftment, one cannot conclude that the used regimen has no impact on the in-vivo growth of MV4;11 AML cells (which, for example, are likely to have a much higher proliferation rate than CD34+ HSCs). In fact, another study (Saland et al., 2015) has documented significant differences in the growth of xenografted MV4;11 cells following irradiation vs. busulfan (20 mg/kg). Although the use of NSG and a lower busulfan dose does not allow for direct conclusions for the present study, these results illustrate how sensitive xenograft models are to conditioning regimens. As most prominent difference, after busulfan conditioning the MV4;11 model turned out to be much delayed compared to the original study, which might be explained by insufficient immunosuppression, toxicity of busulfan or its metabolites on MV4;11 cells, or other experimental differences (which the authors partially summarize in their Discussion). Beyond reducing the overall aggressiveness of the model (and, potentially, promoting a high incidence of xenograft-derived solid tumors, which has been observed by the authors), these altered experimental conditions may also impact the in-vivo responsiveness of MV4;11 cells to I-BET151, which should be discussed. This seems particularly important, since busulfan conditioning is still quite uncommon and the used regimen (30 mg/kg 24 hrs prior to injection) has neither been tested in MV4;11 cells, nor has it been evaluated regarding its impact on treatment responsiveness.

Beyond changing the conditioning protocol, the authors (based on general toxicity studies) also decided to reduce the I-BET151 dose from 30 mg/kg to 20 mg/kg. While this seems reasonable for the specific experiment, the cause for the observed toxicity remains unclear and may (again) be related to the conditioning regimen. Most importantly, the use of a different dosing regimen on its own provides a possible explanation for reduced drug effects in the MV4;11 xenograft model. In fact, in recent work the authors of the original study show that reducing I-BET151 to 10 mg/kg has no impact on survival in this model (Gilan et al., Figure 3). Overall, it cannot be assumed that the drug batch was simply more active and, therefore, needed to be reduced to achieve comparable in-vivo activities. While differences in drug batches should always be considered, the reduced tolerability might be due to the used conditioning regimen, and the differences in survival might simply be a consequence of the reduced I-BET151 dose used in this study.

Overall, changes in the conditioning regimen and the use of a lower dose of I-BET151 may substantially alter the in-vivo response of MV4;11 cells and provide (independent) possible explanations for the observed reduction in in-vivo drug activity. In light of these concerns, it seems very important to not give the wrong impression that the xenograft experiments of the original study could not be reproduced, which would easily be the take-home-message after a "superficial read" of the current manuscript. To avoid this impression, the authors should clearly highlight the most relevant deviations from the original study (i.e. the alternative conditioning regimen and the reduced I-BET151 dose) in the Abstract. Moreover, these experimental differences should also be discussed more extensively in the Discussion. In the current Discussion, the authors first raise some uncertainties due to missing experimental details, then summarize consistent conditions, and then transition into various possible differences that are hard to control for (which are certainly relevant; differences in MV4;11 culture conditions such as different FBS batches could be added). However, for a comprehensive discussion it seems very adequate to first summarize consistent conditions, then highlight obvious and important differences in experimental design (i.e. different conditioning and different drug dose), and only then discuss additional factors that are not easy to control for.

In summary, the design and execution of this replication study is of high quality, and all necessary changes to the original protocol have been implemented in the best possible way. However, the changes in the design of the xenograft study are substantial and might fully explain the observed differences in in-vivo effects of I-BET151. Due to these differences, the results of the performed xenograft experiments are not fully conclusive with respect to the reproduction of the original study, which should be highlighted more clearly.

---

## [Author Response]

*The following issues need to be addressed before acceptance, as outlined below:*

*1) The Abstract and text needs to make clear that (and how) the* in vivo *protocol employed deviated from the original protocol and acknowledge how this might alter the outcome. It seems very important to not give the wrong impression that the xenograft experiments of the original study could not be reproduced, which would easily be the take-home-message after a "superficial read" of the current manuscript. To avoid this impression, the authors should clearly highlight the most relevant deviations from the original study (i.e. the alternative conditioning regimen and the reduced I-BET151 dose) in the Abstract.*

We have revised the Abstract to reflect the most relevant deviations between this replication attempt and the original study. We have also included these deviations and expanded discussion on how they might alter the outcome in the main text.

*2) Complications to interpretation due to other animal health issues should be clearly described. Here, it is not necessary to describe all potential issues in detail in the main text (although perhaps in supplement), but to address the following possibility, raised by reviewer #1. "The fact that the reduction in disease burden following IBET treatment closely paralleled the findings reported in the original study yet a survival benefit was not observed here raises the possibility that leukaemia related mortality was not the primary cause for the mortality observed in at least some animals in the IBET treatment arm."*

We have revised the manuscript to include many of the potential areas of clarification and issues raised in the reviews. Specifically, for the possibility raised by reviewer #1, we included this possibility in an additional paragraph after describing the results of the in vivo experiment.

*Specific examples of places where clarification is in order are given below, as well as some additional suggestions, which have been extracted from the reviews:*

*Reviewer #1:*

*From a reader's perspective there is too much emphasis on the statistical testing performed and not enough discussion about the results and how variations to the original study should be interpreted.*

*For experiments 1) and 2) the general results obtained very much led to the same conclusion. In experiment 1) however, although there was, as previously, a marked difference in sensitivity between MV4;11 (IC50 9 nM) and K562 (unmeasurable), the meta-analysis demonstrated poor reproducibility between the concentrations in the original and reproduced experiments for MV4:11 (IC50 of 26 nM previously).*

*The main area of discrepancy relates to 3) the* in vivo *mouse experiment and specifically the efficacy study assessing overall survival. There are a number of methodological aspects of this study that differ substantially to the original study and these are not raised in the Abstract nor discussed adequately in the manuscript. Therapeutically, the dose of the drug was different (less, at 20 vs 30 mg/kg) as was the duration of therapy, with treatment starting at d35 instead of d21 as in the previous study. As disease activity was determined by examination of peripheral blood only and we know that this xenograft model gives disseminated disease, one interpretation could be that the results differed as a lower dose was used in the face of an increased disease bulk. Most importantly technically, the conditioning regime used was markedly different. 30mg/kg of Busulfan is a very different approach to the more conventional strategy of sub-lethal irradiation performed in the original study. The conditioning regime can dramatically alter engraftment kinetics and also the natural history of the disease. This is reflected in this particular case by the fact that one quarter of the mice on this study developed 'solid malignancies', which I assume to be extramedullary leukaemia. Furthermore 3/12 mice treated with I-BET developed gastrointestinal bloating and (whilst not explicitly described) presumably died for reasons that were independent of the leukaemia burden. This is alluded to by the authors when they state that "animal health issues were observed beyond the anticipated disease presentation of this model" but could be stressed more. These important issues impact significantly on an efficacy study that is essentially measuring all cause mortality. The fact that the reduction in disease burden following IBET treatment closely paralleled the findings reported in the original study yet a survival benefit was not observed here raises the possibility that leukaemia related mortality was not the primary cause for the mortality observed in at least some animals in the IBET treatment arm.*

*All of the above differences would also explain the apparently contradictory finding that, although survival was not impacted, disease bulk was consistently less in the IBET treated arm. A significant issue with the manuscript as currently presented is that the reader is left unaware of these critical details when reading the reproduction. It is essential to communicate this both in the Abstract, in the Results section and when discussing the discrepant results.*

We agree that there are many aspects a reader needs to consider when interpreting this study, just like any individual study. We have revised the manuscript to include the points raised here, specifically including discussion on the deviation in dose and conditioning used and the impact this could potentially have had on engraftment kinetics, disease progression, and determining survival benefit. We have also clarified in the text that the gastrointestinal bloating was observed at the time of sacrifice and expanding discussion about the solid tumors, which as this review suggests is likely extramedullary disease.

*The results from the MTD study show that the mice in the 30mg/kg arm had the greatest variation in starting weight amongst all the cohorts. There is a 25% variation in weight between the heaviest and lightest mouse in this cohort whereas the starting weight of the mice in the other cohort's vary very little. The initial weight of the mice can contribute substantially to the observed toxicity of any intervention especially conditioning. This variation may be one of the reasons why more toxicity was observed in this cohort.*

We agree that variation in the initial weight of the mice could be a reason for observed toxicity, but in the case of the 30 mg/kg cohort, the lightest mouse did not have any observed weight loss throughout the course of the MTD study or have a health monitoring score that justified euthanasia. The remaining four mice had weights with variation similar to the other cohorts; however two of those mice had to be euthanized, which defined the MTD for the study.

*Reviewer #2:*

*The summary assessment of each experiment is not consistently written it states 'did not find a statistically significant' but the converse is not stated, i.e. 'we did see a statistically significant'. The summary needs to be consistent. This replication study has been powered from a statistical perspective so this needs to be clear in the Abstract.*

We have revised the Abstract to provide a balanced summary assessment. We included ‘statistically significant’ for the *BCL2* expression analysis, since we followed through on the confirmatory analysis that was powered from a statistical perspective. However, we did not revise the analysis of IC_50_ values since this was an exploratory analysis.

*Reviewer #3:*

*Most experiments reproduce key results of the original study, with the exception of in-vivo treatment studies in the MV4;11 AML xenograft model, where the authors only reproduce the impact on disease burden while impact on overall survival has been observed. For the interpretation of these results it should be noted that the replication study diverged from the original work in several experimental details that might be of relevance.*

*Overall, the following major points should be considered prior to publication:*

*In the replication of MV4;11 xenograft experiments several experimental conditions deviate from the original study. Most likely due to a lack of access to an irradiation unit, the authors conditioned NOD/SCID mice with a single dose of busulfan (30 mg/kg i.p. 24 hrs prior to MV4;11 transplantation), which has been reported to lead to comparable engraftment of human CD34+ cord blood cells. Apart from the difference in cell type, the original study by Robert-Richard et al. mainly used other dosing regimens (35 mg/kg 36 hrs prior to injection; or 2 x 25 mg 48/24 hrs prior to injection), which they found to be less toxic. While under these conditions, busulfan and irradiation performed similar in human CD34+ HSC engraftment, one cannot conclude that the used regimen has no impact on the in-vivo growth of MV4;11 AML cells (which, for example, are likely to have a much higher proliferation rate than CD34+ HSCs). In fact, another study (Saland et al., 2015) has documented significant differences in the growth of xenografted MV4;11 cells following irradiation vs. busulfan (20 mg/kg). Although the use of NSG and a lower busulfan dose does not allow for direct conclusions for the present study, these results illustrate how sensitive xenograft models are to conditioning regimens. As most prominent difference, after busulfan conditioning the MV4;11 model turned out to be much delayed compared to the original study, which might be explained by insufficient immunosuppression, toxicity of busulfan or its metabolites on MV4;11 cells, or other experimental differences (which the authors partially summarize in their Discussion). Beyond reducing the overall aggressiveness of the model (and, potentially, promoting a high incidence of xenograft-derived solid tumors, which has been observed by the authors), these altered experimental conditions may also impact the in-vivo responsiveness of MV4;11 cells to I-BET151, which should be discussed. This seems particularly important, since busulfan conditioning is still quite uncommon and the used regimen (30 mg/kg 24 hrs prior to injection) has neither been tested in MV4;11 cells, nor has it been evaluated regarding its impact on treatment responsiveness.*

We have revised the manuscript to include the points raised in here, specifically including discussion on the deviation in the conditioning used and the impact this could potentially have had on engraftment kinetics, disease progression, and responsiveness of the MV4;11 cells to I-BET151, especially considering the impact busulfan has on this model is not fully understood.

*Beyond changing the conditioning protocol, the authors (based on general toxicity studies) also decided to reduce the I-BET151 dose from 30 mg/kg to 20 mg/kg. While this seems reasonable for the specific experiment, the cause for the observed toxicity remains unclear and may (again) be related to the conditioning regimen. Most importantly, the use of a different dosing regimen on its own provides a possible explanation for reduced drug effects in the MV4;11 xenograft model. In fact, in recent work the authors of the original study show that reducing I-BET151 to 10 mg/kg has no impact on survival in this model (Gilan et al., Figure 3). Overall, it cannot be assumed that the drug batch was simply more active and, therefore, needed to be reduced to achieve comparable in-vivo activities. While differences in drug batches should always be considered, the reduced tolerability might be due to the used conditioning regimen, and the differences in survival might simply be a consequence of the reduced I-BET151 dose used in this study.*

We agree that changing the dose of I-BET151 could impact the efficacy of the drug independent of the impact other experimental factors might have as well. We have revised the manuscript to highlight this and discuss it in context of other work by authors of the original study who recently reported I-BET151 administered at 10 mg/kg had no impact on survival in a MLL-AF9 leukemia mouse model (Gilan et al., 2016), while I-BET151 administered at 15 mg/kg did provide a survival benefit in a mouse model of NPM1c AML, albeit with a smaller effect size (Dawson et al., 2014).

*Overall, changes in the conditioning regimen and the use of a lower dose of I-BET151 may substantially alter the in-vivo response of MV4;11 cells and provide (independent) possible explanations for the observed reduction in in-vivo drug activity. In light of these concerns, it seems very important to not give the wrong impression that the xenograft experiments of the original study could not be reproduced, which would easily be the take-home-message after a "superficial read" of the current manuscript. To avoid this impression, the authors should clearly highlight the most relevant deviations from the original study (i.e. the alternative conditioning regimen and the reduced I-BET151 dose) in the Abstract. Moreover, these experimental differences should also be discussed more extensively in the Discussion. In the current Discussion, the authors first raise some uncertainties due to missing experimental details, then summarize consistent conditions, and then transition into various possible differences that are hard to control for (which are certainly relevant; differences in MV4;11 culture conditions such as different FBS batches could be added). However, for a comprehensive discussion it seems very adequate to first summarize consistent conditions, then highlight obvious and important differences in experimental design (i.e. different conditioning and different drug dose), and only then discuss additional factors that are not easy to control for.*

We have revised the Abstract to reflect the most relevant deviations between this replication attempt and the original study. We have also reordered the experimental differences as suggested and included these deviations and expanded discussion on how they might alter the outcome in the main text.

In summary, the design and execution of this replication study is of high quality, and all necessary changes to the original protocol have been implemented in the best possible way. However, the changes in the design of the xenograft study are substantial and might fully explain the observed differences in in-vivo effects of I-BET151. Due to these differences, the results of the performed xenograft experiments are not fully conclusive with respect to the reproduction of the original study, which should be highlighted more clearly.

We agree that there are many aspects a reader needs to consider when interpreting this study, just like any individual study and have revised the manuscript to address the points raised in these reviews.